# Dual Effect of Nanomaterials on Germination and Seedling Growth: Stimulation vs. Phytotoxicity

**DOI:** 10.3390/plants9121745

**Published:** 2020-12-10

**Authors:** Réka Szőllősi, Árpád Molnár, Selahattin Kondak, Zsuzsanna Kolbert

**Affiliations:** Department of Plant Biology, University of Szeged, H-6726 Szeged, Hungary; molnara@bio.u-szeged.hu (Á.M.); selahattinkondak@gmail.com (S.K.); kolzsu@bio.u-szeged.hu (Z.K.)

**Keywords:** nanomaterials, seed germination, root elongation, seed priming, phytotoxicity

## Abstract

Due to recent active research, a large amount of data has been accumulated regarding the effects of different nanomaterials (mainly metal oxide nanoparticles, carbon nanotubes, chitosan nanoparticles) on different plant species. Most studies have focused on seed germination and early seedling development, presumably due to the simplicity of these experimental systems. Depending mostly on size and concentration, nanomaterials can exert both positive and negative effects on germination and seedling development during normal and stress conditions, thus some research has evaluated the phytotoxic effects of nanomaterials and the physiological and molecular processes behind them, while other works have highlighted the favorable seed priming effects. This review aims to systematize and discuss research data regarding the effect of nanomaterials on germination and seedling growth in order to provide state-of-the-art knowledge about this fast developing research area.

## 1. Introduction

The term “nanomaterial” (NM) refers to a material with one dimension under 100 nm [1,2,3,4]. With the development of nanotechnology, the use of nanomaterials is seeing an unprecedented increase, and studies are needed to focus on the effects of nanomaterials in all living organisms, especially sessile plants that cannot avoid these kinds of external factors. It is also essential to evaluate the possible hazards of nanomaterials in the environment, as well as in plants, animals and humans, because of their increasing emissions. For example, the global output of ZnO nanoparticles (NPs), which are widely applied, is between 550 and 5550 tons per year, a value that is approximately 10–100 times higher than that of other NMs [5]. 

Most of the physico-chemical properties of NMs vary depending on shape, size, surface area, surface/volume ratio, chemical behavior, particle charge, production method, coating, and so on, as has previously been described in detail [6,7]. Changes in NM synthesis can lead to magnetic properties in NMs that can be useful in medical processes [8]. NMs are often modified with oxides or other molecules to increase conductivity and help avoid aggregations of NMs, and this has a significant impact on NM behavior [9]. According to their dimensionality, NMs can be divided into four categories: 0D NMs, where the electrons are confined in all three dimensions (e.g., quantum dots (QDs)); 1D NMs, where the electrons can move in one dimension (e.g., quantum wires and nanofibers); 2D NMs, where the electrons can move in two dimensions (mostly nanofilms and nanosheets); and 3D NMs, which are usually made of other NMs and allow electrons to move freely in all three dimensions [10]. The morphology of NMs is diverse, from nanocubes and nanopyramids to nanowires and nanozigzags. NMs can be composed of a single material, or they can be used as composites. The uniformity of NMs, especially nanoparticles, can be isometric if all particles are roughly the same size or inhomogeneous. This attribute will influence the behavior of agglomerated NPs in solution [9]. Based on these features and previously published reports, we suppose that each NM may have specific impacts (either beneficial or toxic) on living organisms like plants; therefore, NMs must be characterized in every case study. 

Classification of NMs is depicted in Figure 1. The difference between properties of NM groups is significant, and all main NM groups (i.e., carbon-based, metal-based NMs, quantum dots, dendrimers and nanocomposites) will be described later in detail. 

Nanofibers (NFs) are a relatively new member of one-dimensional nanomaterial, like nanotubes or nanowires. Their large surface areas, high tensile strength and porosity make them interesting to the industry. NFs are made cost effectively through electrospinning, a method devised in the USA in 1902 [11]. With recent advances in electrospinning, both synthetic and natural polymers are usable in the production of NFs. It is worth mentioning that, besides polymeric NFs, ceramic and metal oxide NFs can also be created. The morphology, diameter (from 10 to several hundred nanometers), chemical composition and surface modifications of NFs can be modified [12]. New advances have made nanofibers a promising material for medical and cosmetic applications such as tissue or organ repair [13]. 

Nanoclays and organoclays are a type of silicate with general thickness of 50–200 nm. They are produced utilizing the hydrophilic nature of clay molecules via an ion exchange reaction in the aqueous or solid state. During the reaction, the gap between clay layers is widened, enabling organic cation molecule integration between the layers. The surface of the clay sheets also changes from hydrophilic to hydrophobe. These NMs have antimicrobial and toxin absorption capabilities, making them ideal for applications in food industry (food packaging) [14]. 

Emulsions are defined as the dispersion of two immiscible liquids [15]. Nanoemulsions are made in a viscous liquid via the dispersion of polymers, droplets or other solid materials, and is referred to as dispersed or discontinuous phase. Physical properties of these liquids, such as viscosity, phase behavior and density are influenced by oil phase components [16]. These stable colloids are usually used in the food industry to develop biodegradable food packaging, to increase the shelf lives of foods and as a decontaminant for equipment [17]. 

The word “nanoparticle” (NP) was coined recently and usually refers to manufactured anthropogenic NPs. By definition, NPs have at least two dimensions between 1–100 nm, which include naturally occurring particles (e.g., dust, aerial particles, colloids, etc.). In nature, these structures are common and have been identified in glacial ice cores and the Cretaceous–Tertiary boundary layer [18,19,20]. In samples of this layer from Italy, magnetic iron materials, hematite and silicate has been found with sizes between 16–27 nm [19]. The origins of different NPs are summarized in Figure 2. 

In this study, due to their relatively fresh appearance and widespread use compared to natural NPs, anthropogenic NPs are discussed. Manufactured NPs include organic forms, such as carbon structures, polymers, dendrimers, lyposomes and micelles, while inorganic forms include metal oxides, metals and quantum dots (QDs). Compared to their bulk forms, their physical and chemical characteristics differ greatly [22], as their unique nanostructures have excellent properties. 

NPs are widely used across the industries and the rapid market increase in the previous decade was predicted [23,24]. The growth of their production and disposal has enhanced the possibility of NPs being released into the environment and coming into contact with living organisms, such as plants [22,25,26]. Studies dealing with the effects of NMs establish that many factors can have an impact on the exact outcome of the NM–plant interactions, including the plant species, the size of the applied NMs, the duration or existence of pre-cultivation (e.g., seed priming), the concentration and span of NM exposure or the growing conditions, namely the germination test performed in Petri dishes or hydroponics or a pot experiment using soil. To date, it has been well documented by reviews and case studies that how several NMs, mainly metallic NPs, may positively or negatively influence biomass, the photosynthetic activity or the yield of adult plants, but there remains a great lack of knowledge concerning the early developmental stage, i.e., seed germination and early seedling growth [27,28,29]. The uptake of NMs by mature (adult) plants has been documented by several studies. A large number of studies reported ZnO NPs entering plant tissues or cells [22,30,31,32]. In case of carbon-based NPs, single-walled carbon nanotubes (SWCNTs) have been identified inside plant tissues [33,34]. A recent report has reviewed in detail how carbon-based NMs can be uptaken by root or shoot (due to foliar application), translocated via the vascular tissues and affect plant growth or modulate stress tolerance (summarized by [35]). However, most NMs and especially NPs have been reported in studies as materials entering plant tissues (overviewed by [36]), thus further research has to be conducted to reach a conclusion on this topic, due to the diversity of NMs in terms of their size and morphology. 

Since seed germination is the first step and the most sensitive stage of higher plants’ ontogenesis, studying the effects of NMs during this phase seems to be very informative for researchers and agronomists. Since 2000, a growing number of reports (>1000, Science Direct, [37]) have analyzed the impact of metal-containing nanosized materials on seed germination, especially the positive/negative effects of NPs which are broadly applied in agriculture, electronics, cosmetics, medicine, etc. (overviewed by [38,39,40]). 

During seed germination, the NMs (NPs or QDs) first have to penetrate the seed coat which generally contains sclerenchyma, namely sclereids, and due to its physical–chemical integrity, it can act as a barrier for the NMs [41]. Some reports suggest that NMs use the intercellular spaces of the tissues or create new pores mainly through the upregulation of aquaporin production (discussed by [38,42]). Another recent study has also confirmed that some metal oxide NPs (ZnO and TiO_2_) can get through the seed coat and provoke the embryonic differentiation via stimulating the enzymes which are involved in interrupting seed dormancy [38]. When the radicle of the new progeny emerges, the developing tissues of the root apex get in touch with the NMs which may enter the rhizodermis via apoplastic transport, endocytosis or other carriers, then within the root they flow toward the vascular cylinder using symplastic pathways and are translocated to other progressing plant parts (discussed by [28,43]). 

Naturally, it is also important to know whether the uptaken NMs are biotransformed into their ionic form, and whether they can be detected in plant tissues in nano-form, as it was reviewed by [27,28,43]. 

Up to now, in most experiments, germination tests have been executed in Petri dishes applying NM-containing agar or wet filter paper, and it has been assessed that the exposition to NMs may affect the efficiency of germination (with parameters like germination percentage, mean germination time or seedling vigor index) and the early plant growth (radicle/root length and plumule/shoot length, see [44]), as we will discuss it in this review. 

## 2. Effects of Nanomaterials on Seed Germination and Seedling Growth

### 2.1. Concentration-Dependent Effects of CNMs on Seed Germination and Seedling Growth 

The first carbon-based nanomaterial was a 60-carbon atom fullerene, discovered in 1985 [45]. In 1991, a fullerene product, carbon nanotubes (CNTs), were first manufactured [46]. The synthesis of carbon nanotubes continued with multi-walled carbon nanotubes (MWCNTs) with 10 µm length and 5–40 nm diameter. With added cobalt-nickel catalyst, the production of single-walled carbon nanotubes (SWCNTs) was achieved. The strength to weight ratio of SWCNTs is 460 times larger compared to steel. Nowadays, the types of carbon NMs are numerous: fullerenes and fullerene cages, SWCNTs and MWCNTs, cup-stacked carbon nanotubes, graphene sheets, etc. [46,47,48,49]. Most carbon NPs are hydrophobic, leading to an aggregation or precipitation in aqueous solutions. Due to the large differences in morphology and chemical properties, individual carbon-based NPs are a diversified group, with large industrial usage among NMs. Due to their large usage, there is a growing concern that CNTs behave similarly to asbestos and are harmful to human health [50]. 

#### 2.1.1. Carbon Nanotubes (CNTs)

In the early work of Lin and Xing [51], the influence of MWCNTs on seed germination and seedling growth of six different crop species such as radish (*Raphanus sativus*), rapeseed (*Brassica napus*), ryegrass (*Secale cereale*), lettuce (*Lactuca sativa*), maize (*Zea mays*) and cucumber (*Cucumis sativus*) was evaluated. Germination was not affected by MWCNTs in either of the examined species but seedling root growth was enhanced in ryegrass and maize [51]. Similarly, MWCNTs (1000 mg L^−1^) had no effect on the germination process of zucchini (*Cucurbita pepo*) and carrot (*Daucus carota*) [51,52]. However, MWCNTs at the concentration range of 10–40 mg L^−1^ notably enhanced seed germination and seedling growth in tomato (*Solanum lycopersicum*) [53] and the promoting effect of MWCNTs was supposed to be due to their capability of penetrating the seed coat and promoting water uptake. MWCNTs stimulated cell growth in tobacco BY2 cell suspension, which was accompanied by the upregulation of genes involved in cell division (CycB)/cell wall formation (NtLRX1, extensin) and water transport (NtPIP1, aquaporin), providing a molecular explanation for the growth-inducing effect of MWCNTs [54]. The work of Cañas et al. [55] compared the effects of nonfunctionalized and functionalized (with poly-3-aminobenzenesulfonic acid) SWCNTs and observed that the effect was dependent on the plant species, since the root growth was not affected in cabbage (*Brassica oleracea*) and carrot, but was inhibited in tomato, while it was promoted in onion (*Allium cepa*) and cucumber seedlings. It was also observed that nonfunctionalized CNTs affect root length more seriously than in their functionalized form. Using tobacco BY2 cells, Liu et al. [56] convincingly showed that SWCNTs are able to penetrate through the plant cell wall and plasma membrane, supporting the observed effects on seedling growth, which is probably due to their size (1–15 nm), often smaller compared to MWCNTs [57]. Several further reports have indicated that seed germination and/or seedling growth is induced by MWCNTs and SWCNTs in plant species like the tomato, Indian mustard (*Brassica juncea*), onion, radish (*Raphanus sativus*), turnip (*Brassica rapa*), sage (*Salvia officinalis*), pepper (*Capsicum annuum*), tall fescue (*Festuca arundinacea*), wheat (*Triticum aestivum*), maize, peanut (*Arachis hypogaea*), garlic (*Allium sativum*), rice (*Oryza sativa*), barley (*Hordeum vulgare*) soybean (*Glycine max*), switchgrass (*Panicum virgatum*), and gram (*Cicer arietinum*) [58,59,60,61,62,63,64,65,66], as reviewed by [67] and [68]. The concentration-dependent effect of MWCNTs, Fe-filled carbon nanotubes (Fe-CNTs), and Fe–Co-filled carbon nanotubes (FeCo-CNTs) was compared in the study of Hao et al. [69], where the seedling’s root length was increased by the low concentrations. However, auxin (IAA) content in rice roots and shoots decreased upon the exposure to all of the three CNTs at all concentrations. Additionally, CNT treatment resulted in decreased levels of other phytohormones including gibberellin (GA1+3), cytokinin (IPA), jasmonic acid (JA), brassinolide (BR), and abscisic acid (ABA). These changes in hormonal status may contribute to the negative effects of the examined CNTs [69]. Moreover, in case of *Hyosciamus niger*, MWCNTs decreased the germination percentage and increased the germination time and the early seedling growth was decreased as well [70]. MWCNT treatment caused oxidative stress, which was supported by the elevation of lipid peroxidation, electrolyte leakage, H_2_O_2_ and also by the activation of the antioxidant defense [70]. These results were supplemented by Khalifa [71], who observed that the toxic effects of high MWCNT doses (100 and 200 μg μL^−1^) are associated with the binding of MWCNTs to genomic DNA. 

In contrast, the application of CNTs increased the germination rate of *P. virgatum* seeds and speeded up the germination process of sorghum (*Sorghum bicolor*) seeds as well as promoted seedling growth [72]. A similar positive effect of CNTs on tomato seedling growth was observed by [73] where modified antioxidant response and the increased production of antioxidant compounds were found. Seed priming with MWCNTs functionalized with carboxylic acid (MWCNT–COOH) proved to be effective in improving seed germination and seedling vigor in buffaloberry (*Shepherdia canadensis*) and green alder (*Alnus viridis*) [74]. An important novel finding of this study was that the cessation of both embryo and seed coat dormancy was associated with the remodeling of C18:3-enriched fatty acids in seed membrane lipid molecular species, suggesting that MWCNTs functionalized with carboxylic acids modulates cell membrane lipid metabolism [74]. The concentration-dependent effect of CNTs (and graphene) was further supported in tomato seedlings where seed priming had no effect on the germination process, but increased root biomass and activated antioxidants (ascorbic acid, phenols, flavonoids, superoxide dismutase (SOD), catalase (CAT), GPX (glutathione peroxidase), etc.) [75]. 

Recently, the stress modulating effect of CNTs was investigated by several research groups. It was reported that MWCNT treatment aggravated the negative effects of cadmium (Cd) on root elongation, lateral root and root hair formation, root and shoot biomass formation and Cd accumulation was induced by MWCNTs [76]. In case of drought-stressed *Glycine max* seeds, however, SWCNTs improved germination and seedling growth by reducing lipid peroxidation and H_2_O_2_ content but increasing ascorbic acid (AsA) content and SOD, CAT, peroxidase (POD) activities suggesting that SWCNTs may play an important role in the improvement of antioxidant capacity of soybean seedlings under drought stress [77]. In the work of Baz et al. [78], twenty-seven varieties of *L. sativa* (lettuce) were compared for their sensitivity to salt stress, and the seeds were pre-treated with CNTs. Pre-treatment with CNPs significantly improved seed germination in the case of salt exposure (150 mM NaCl), and high temperature; however, different lettuce varieties exhibited distinct responses to nanoparticle treatments drawing attention to the genotype-dependent effect of CNTs [78]. 

The large amount of experimental data indicates that the effect of MWCNTs and SWCNTs on seed germination and seedling growth shows concentration dependence, dependence on the plant species, on the plant genotype and also on the treatment conditions. Therefore, the optimal circumstances and growth-promoting concentrations are recommended to be experimentally verified before practical application.

#### 2.1.2. Carbon Nanodots (CDs)

As for the effect of carbon dots (CDs) on seed germination and seedling growth, there are only few results available. The first study was conducted on *Zea mays* plants where high doses of CDs (1000 and 2000 mg L^−1^) led to decreased root and shoot biomass due to H_2_O_2_ accumulation and intensified lipid peroxidation. Additionally, CD exposure activated antioxidant enzymes like CAT, APX, GPX and SOD. CDs were visualized in root-cap cells, cortex cells and vascular bundle of roots and also in leaf mesophyll, indicating the effective absorption and translocation of CDs in maize. Interestingly, the excretion of CDs from leaf blade was also observed [79]. Using a wide concentration range of CDs (0.02–0.12 mg mL^−1^) for treating mung bean (*Vigna radiata*) sprouts, a concentration-dependent effect was observed since the sprouts showed root and stem elongation, increased biomass production and carbohydrate content as the effect of low CD doses. Additionally, CDs enhanced RuBisCO activity and chlorophyll content in the sprouts, suggesting improved photosynthesis [80]. In another study, *V. radiata* sprouts were cultivated in the presence of N-doped C-dots (N-CDs) and a significant enhancement in the sprouts’ yield was observed compared to the aqueous control [81], indicating the effectiveness of N-CDs as a nitrogen nanofertilizer. Qian et al. [82] compared the in planta distribution and the effects of three types of CDs (bared CDs, CD-PEI (modified by polyethylenimine), and CD-PAA (modified by polyacrylic acid)] on growth of *C. pepo* seedlings. It was found that all three types of CDs triggered the antioxidant defense systems (SOD, POD, CAT) [82]. The available literature has recently been reviewed by [83]. Furthermore, in a comparative study, the most significant promoting effect of functional CDs (FCNs), possessing the largest number of functional groups and a small size, on the growth of *Arabidopsis thaliana* seedlings, was observed. The remarkable effect of FCNs may be due to their perfect aqueous dispersity, nutrient adsorption capacity and bioaffinity, as suggested by the authors [84]. 

#### 2.1.3. Carbon Nanohorns (CNHs)

Carbon nanohorns (CNHs) are a promising carbon-based nanosized material with special characteristics. Unlike carbon nanotubes, CNHs are uniform in size and can be well dispersed in solvents. Moreover, they can be synthesized in large quantities without any catalyst [65]. The germination and growth-promoting effect of single-walled carbon nanohorns (SWCNHs) were evidenced at the physiological, cellular and genetic levels in the study of Lahiani et al. [64] using barley, maize, soybean, rice, switchgrass and tomato seeds. The germination of barley and soybean showed only a slight response to SWCNHs, while the germination percentage of corn, rice, tomato and switchgrass significantly improved under the effects of all three SWCNHs concentrations (25, 50 or 100 mg L^−1^) compared to the control [64]. As for the seedling development, SWCNHs exerted inducing effects on shoot and root length, leaf number, as well as fresh and dry weights, however, the effects proved to be concentration-dependent and were dependent on the plant species. This study also confirmed that the growth of tobacco cells is induced by SWCNHs and that SWCNHs are able to affect the expression of a number of tomato genes that are involved in stress responses, cellular responses and metabolic processes [64]. Recently, the effect of SWCNHs on the root system growth of *Arabidopsis thaliana* seedlings was evaluated at the molecular and metabolic levels [85]. A low concentration of SWCNHs (0.1 mg L^−1^) promoted primary root (PR) elongation and lateral root (LR) formation, as well as increased the lengths of the meristematic and elongation zones. It was further confirmed that SWCNHs enhanced stem cell niche activity, meristematic cell division potential and the auxin level and signaling of *Arabidopsis* root apex. Metabolomics supported by transcriptomic data revealed that SWCNHs reprogrammed carbon/nitrogen metabolism and increased the levels of secondary metabolites (e.g., serotonin, hypoxanthine, adenine). These data provide insight into the molecular basis of the growth promoting effect of SWCNHs [85]. 

#### 2.1.4. Fullerenes and Fullerols

In the work of Liu et al. [86], the water-soluble fullerene malonic acid derivative (FMAD) inhibited the root and hypocotyl elongation of *Arabidopsis* seedlings in a concentration-dependent manner, although the germination capacity was not affected, possibly due to the protective effect of the seed coat. The observed root-shortening effect of FMAD is associated with the disruption of cell division, microtubule arrangement, auxin levels and with intracellular ROS (reactive oxygen species) accumulation. In contrast, polyhydroxy fullerene (PHF or fullerol) treatment at high concentrations (100,000 and 200,000 mg L^−1^) exerted a significant positive effect on the root and hypocotyl elongation of *Arabidopsis* seedlings [87]. In addition, PHF promotes the elongation of barley roots due to the enhancement of their longitudinal extensibility in the elongation root zone [88]. Additionally, in the presence of a stressor such as UV-B radiation, salt stress or the presence of a high salicylic acid dosage, PHF exerted a more pronounced effect on root growth. PHF protected seedlings from oxidative damage induced by UV-B irradiation, suggesting that PHF is able to enhance growth due to its ROS scavenging capacity [88]. Xiong et al. [89] applied seed priming with PHF and observed a significant inducing effect on seed germination in the case of polyethylene glycol (PEG)-triggered osmotic stress. Additionally, the foliar application of PHF led to an increment in shoot dry weight and photosynthesis in rapeseed (*Brassica napus*) seedlings grown in dried soil. The level of ROS decreased and the content of antioxidants as well as the activities of antioxidant enzymes increased in PHF+drought-treated seedlings compared to seedlings exposed to drought alone. It was also observed that the PHF treatment of drought-stressed seedlings induced an elevated ABA content in the leaves and triggered ABA biosynthesis by downregulating the expression of the ABA catabolic gene CYP707A3. In a recent study, the protein profile of maize seeds during fullerene-influenced germination was examined [90]. Maize seeds showed to have a higher germination rate and faster germination due to the effect of the water-soluble quaternary ammonium salts of iminofullerenes (IFQA). Upon IFQA treatment storage, proteins (e.g., globulin, vicilin-like embryo storage protein) were downregulated and proteins involved in energy production (e.g., glyceraldehyde-3-phosphate dehydrogenase 2) and sugar metabolism (e.g., UDP-glucose 6-dehydrogenase isoform 2) were upregulated, explaining a faster germination.

#### 2.1.5. Graphene and Graphene Oxide (GO)

Due to its special characteristics, graphene has great potential in industrial, biomedical and agricultural applications. Therefore, the effects of graphene and graphene oxide (GO) on the germination and seedling growth were evaluated by several authors. For instance, Nair et al. [91] revealed that *Oryza sativa* seedlings, germinated in the presence of graphene, showed better viability and growth compared to untreated seedlings. Similarly, the germination capacity of tomato seeds was increased by powdered graphene possibly due to the ability of graphene to improve water uptake via the seed coat [92]. On the other hand, several studies reported that seed germination was delayed and/or inhibited by graphene or GO application. For instance, *O. sativa* seed germination was delayed by increasing graphene concentrations (5–200 mg L^−1^, [93]). In another short-term study, graphene (250, 500, 1000 and 1500 mg L^−1^) significantly improved root elongation, but inhibited root hair development, which may be associated with graphene induced-oxidative stress in the roots of wheat seedlings [94]. In maize seedlings, sulfonated graphene NPs at low concentration (50 mg L^−1^) stimulated growth (plant height, root and shoot biomass), while a high dosage (500 mg L^−1^) exerted a strong inhibitory effect accompanied by Ca^2+^ signaling, ROS production and lipid peroxidation [95]. During the comparison of the effects of GO and amine-modified graphene (G-NH_2_) it was found that at high concentrations (500, 100 or 2000 mg L^−1^), GO inhibited wheat germination and seedling growth, while the same doses of G-NH_2_ exerted positive effects. The electrolite leakage of roots was increased by GO exposure supporting the toxic nature of this nanomaterial type [96]. According to Vochita et al. [97], wheat seed germination was inhibited by a high dosage of GO (2000 mg L^−1^) and a slight reduction in root elongation was also observable at this concentration. Moreover, the increment in chromosomal aberrations and mitotic abnormalities indicates the clastogenic and aneugenic effect of GO in wheat root meristem. Recently, Xu et al. [98] have claimed that 10 nm-sized graphene quantum dots (GQDs) can promote the absorption of water and nutrients by increasing the effective surface areas of the root epidermal (rhizodermal) cells. Their schematic model shows that GQDs directly attach to the surfaces of the plant root cells, growing absorptive area for the ions on the root surface, but there is no information about the mode of penetration and further transport within the root.

The presented examples clearly show that the effect on early plant development depends on the concentration of graphene or GO. Due to its capability in transporting water, graphene improves seed germination; however, elevated doses cause oxidative stress and genotoxicity.

The promoting influences of carbon-based NMs (CNMs) on seed germinations are summarized in Figure 3.

### 2.2. The Influences of Metal-Based NMs on Germination and Seedling Growth 

The effect of metallic nanomaterials (NMs) (including metal and metal oxide NPs, and quantum dots, QDs) on plant development and physiological processes is an intensely researched area, since plants being the first step of food web have a key role in a potential NM contamination. It was demonstrated that, e.g., QDs, known as nanoscale autofluorescent semiconductors, are not only uptaken by plants like *Arabidopsis* but are transferred to its herbivores, as well [99]. Moreover, seed germination, including the emergence of the radicle and the elongation of the primary root, is the most sensitive part of the plant life cycle, therefore both the beneficial and negative effects of metallic NMs can be well tested. 

The use of metals and metal oxides dates back to ancient times, as titanium oxide (TiO_2_) was used as paint in ancient Egypt. TiO_2_ and zinc oxide (ZnO) NPs are widely used across the industry. Their high surface area compared to their weight and volume, high reactivity and high chemical mechanical and heat stability resulted in a diverse use for all metal oxide NPs, especially ZnO. Metal oxide NPs are used in sunscreens, paint and solar cells, laser technology, etc. Interestingly, the yearly production of ZnO NPs is estimated to be 10–100 times larger than other NMs [5]. Zero valent metal NPs are made with the reduction of metal salts, where the reductant type and conditions affect the physical properties of the NPs [100]. Zero valent iron has been used as a detoxifier against nitrates in remediation processes and new research suggests organochloride pesticides as a new remediation target [101]. Silver NPs are widespread in the industry [102] and used in air filters, washing machines, baby products and wound dressing. Both the metallic silver NPs and ionic silver have been used extensively. Nanosized silver is reactive in aqueous solutions, resulting in the short half-life of the active form. This resulted in the absorption of silver NPs on other macroparticles, resulting in a stable colloidal form which is still referred to as nano silver by the manufacturers [103,104]. In medical applications, gold nanocolloids are not rare and nanoparticulate gold is used in electronics and as a catalyst. 

#### 2.2.1. Metallic NPs

Similarly to other NMs, metal-containing NPs have been shown to have dual effects in plants, including seed germination. Beneficial influences of elemental metallic NP application were displayed in some crops. The germination of cucumber and lettuce seeds was promoted by solutions containing 62 μg mL^−1^ Au NPs for 7 days [105], and similar results were found in the case of *Pennisetum glaucum* (pearl millet) after soaking the seeds for 2 h in Au NP (20 and 50 μg mL^−1^) ([106]; Table 1).

In addition to soaking, priming seeds with metallic NPs (so-called nanopriming) also seems to be effective. When Almutairi and Alharbi [107] primed watermelon and zucchini seeds for 2 h in Ag NP solution at low concentrations (0.5–2.0 g L^−1^), then germinated the seeds at the same doses, the germination % significantly increased, and root elongation was also promoted. Similarly, Prażak et al. [109] found that Ag NP-primed seeds of bean germinated at a higher rate compared to the control (Table 1). 

Nevertheless, it may occur that metallic NP exposure has no influence on germination and early growth parameters, as it was presented in Ag NP-treated lettuce ([105]; Table 2), *Pinus sylvestris* and *Alnus subcordata,* which were germinated in Ag NP-containing soil ([118], Table 2). In the latter study, seeds were also exposed to Ag NP solutions in Petri dishes, but the germination % and seedling length were negatively affected, which suggests that the character of the growing medium is determinative in the early development of plants under NM application.

Ag NPs, which are mainly used due to their antibacterial activity in medical practice, can be detrimental for germinating seeds and developing plantlets [134]. In the study by Geisler-Lee et al. [134], *Arabidopsis* seeds were germinated in the presence of 20, 40 or 80 nm-sized Ag NPs at 66.84, 133.68, 267.36 and 534.72 mg L^−1^ concentrations, then the elongation rate of the roots was determined. The uptake of Ag NPs by the roots was clearly exhibited, moreover, Ag NP toxicity was shown to be size and concentration dependent. While 80 nm-sized Ag NPs were only deteriorative at higher doses, those of 20 and 40 nm caused severe growth inhibition of the root. All the Ag NP-treated roots had typical brownish root apices, and the NPs were localized in border cells, root cap, columella and columella initials. The researchers supposed that Ag NPs were apoplastically transported through the root tissues. The inhibitory effect of Ag NPs on the germination index was also seen in the case of cucumber ([105]; Table 3). 

Though copper (Cu) is an essential element for plant growth and has a key role in the photosynthesis, we have a few data about the impacts of Cu NPs on seed germination and early seedling development. Lee et al. [135] tested the effect of Cu NPs on mung bean and wheat at relatively high doses (200–1000 mg L^−1^) and both seedling growth and root growth were shown to decrease in a concentration-dependent manner (Table 3). Much lower doses (0.2–1 mg L^−1^) of Cu NPs were used in the study of Hafeez et al. [140] in the germination test executed on wheat (Pakistani wheat cultivar Millat-2011) and only the highest concentration evoked a lower germination percentage. In the experiment of Zuverza-Mena et al. [141], cilantro (*Coriandrum sativum*) seeds (fruits) were germinated in the soil containing 20 and 80 mg kg^−1^ Cu NPs. Germination % was negatively influenced only at the lower concentration, root elongation seemed to be not affected, while shoot length significantly decreased at the higher dose of Cu NPs. From these results, it seems that not only the concentration but the type of growth medium may affect the outcome of a study applying various metallic NPs.

#### 2.2.2. Metal Oxide NPs

Among metallic oxide NMs, ZnO NP is one of the most widely used engineered NMs in the industries, e.g., cosmetics or therapeutics, due to its anticancer and antimicrobial effect (reviewed by [37,142]). It has been documented in numerous plant species (including crops) that the optimal amount of ZnO NPs might have a positive effect on seed germination and seedling growth, also depending on particle size. In germination tests executed in Petri dishes, the germination efficiency expressed as germination % seemed to be induced by ZnO NP application in the case of cucumber [115], mung bean [116], and chili pepper [114]. In addition, ZnO NP application proved to be beneficial in pot and field experiments as well ([112,117]; Table 1). 

At the same time, ZnO NP exposure may also have no effect on germination %, as it was demonstrated in soybean [30] and rice [130] germinated in Petri dishes, in barley [128], bean [132] and in scots pine [118] cultivated in pot experiments (Table 2). Nevertheless, the dual aspects of ZnO NPs were exhibited in case of onion [126], oat [127] and *Brassica* species [32] when germination % and/or primary root elongation were promoted by a low dosage of ZnO NPs but those were inhibited by higher concentrations (Table 2). In the latter study [32], the authors presumed that the ZnO NPs of small size (8 nm) could enter the root cells and be at least partially biotransformed into ionic form. Moreover, ZnO NPs might cause the imbalance of ROS and/or RNS (reactive nitrogen species), resulting in the lower viability of root meristem cells and inhibited root elongation, especially in the sensitive species.

From many studies, it has emerged that ZnO NPs can act as a stressor, inducing a reduction in both seed germination as well as the growth of the radicle and the plumule. Not significant but diminished germination % was observed under ZnO NP exposure in rapeseed [139], wheat and cucumber [130]. Severe inhibitory effects of ZnO NPs were displayed in the case of *Arabidopsis* at all doses [137], while tomato proved to be sensitive at higher (>750–800 mg L^−1^) ZnO NP concentrations ([115,131]; Table 3). 

According to these findings, we may presume that there might be a correlation between the inhibition of root elongation and mitotic disorders in the root tip cells, accompanied by the increase in root diameter [143] or lateral root number [144], which suggests the potential reorientation of root cells like in so-called stress-induced morphogenic responses (SIMR, [145]). Moreover, the tolerance against ZnO NP exposure appears to be related to the constitution of the root cell wall, namely the increase in the amount of lignin, suberin, pectin or callose, as it was presented by Molnár et al. [43]. 

TiO_2_ NPs, which are mainly applied in the cosmetics and paint industry, can differently influence the seed germination process of plants. Its promotive impact was established in the germination tests of *Alyssum homolocarpum, Nigella sativa* and *Salvia mirzayanii* ([111], Table 1). In other experiments executed on onion [125], white mustard [111] and rice [123], lower concentrations (10–100 mg L^−1^, depending on plant species) were found to be stimulative for germination or root growth while higher doses proved to be toxic (Table 2). Absolutely deleterious aspects of TiO_2_ NPs were presented in *Vicia narbonensis* and maize seedlings, where root growth inhibition induced by NPs was due to chromosomal aberrations in root tip meristem cells ([138], Table 3).

Adhikari et al. [146] reported that the germination itself of soybean and chickpea seeds exposed to CuO NPs at 5 up to 2000 mg L^−1^ (<50 nm) was not inhibited probably due to the seed coat but later toxic effects were exhibited in the root elongation in the case of both crops. The roots of both soybean and chickpea seedlings were remarkably shorter compared to the control from 600 mg L^−1^. The negative effects of CuO NPs were also proved in graminaceous crops, rice and maize ([136], Table 3).

Fe_3_O_4_ NPs, which are preferred in biomedicine (e.g., cancer nanotherapy, [39]), were exhibited to have a slight beneficial effect in case of chickpea, since NP-application improved the germination time at all concentrations and induced better shoot growth (Table 2, [122]). At the same time, in the study of Barrena et al. [105], Fe_3_O_4_ NPs proved to inhibit the germination of cucumber and lettuce seeds and reduce the elongation of the primary roots, which was probably due to the small size (7.57 ± 5.6 nm) of the NPs and their potentially easier uptake by the developing roots. Furthermore, excess iron (Fe) can be highly reactive and toxic because of ROS overproduction via the Fenton reaction [147].

#### 2.2.3. Metal Containing Quantum Dots

Quantum dots are photoreactive nanocrystal consisting of a core and a shell. The optical characteristics of the quantum dot is determined by the reactive core, which can be made with metals and semiconductors, for example cadmium selenide (CdSe), zinc selenide (ZnSe), etc. To date, the use of these NMs has been limited to the medical field and new industrial applications such as solar panels or telecommunications [148]. However, in biological systems, QDs have already been used for biolabeling in animal cells [99], and we have only a few data about the impact of these nanocrystals on early plant development. When rice seeds were treated with water-soluble MPA-linked CdSe quantum dots (QDs) at three different concentrations (0.25 mL QDs+1.25 mL H_2_O, 0.5 mL QDs+1 mL H_2_O and 1 mL QDs+0.5 mL H_2_O), the researchers found that there were higher concentrations that blocked germination while the very low concentration of QD solution had no inhibitory effect [149]. Das et al. [150] investigated the effect of the ultra-small size (<5 nm) N-acetyl cysteine (NAC)-coated core–shell CdS: Mn/ZnS QDots (NAC-QDs) on the seed germination and seedling growth of garden pea. The seeds were soaked for 24 h in different concentrations of NAC-QD (2, 5, 10, 20, 40, 60, 80 and 100 μg mL^−1^), then germinated in Petri dishes containing the test solutions for 5 days. Germination % was similar to the control up to 40 μg mL^−1^, but at higher doses, significant decrement occurred with visible signs of root growth inhibition and brownish root tips. 

However, the investigations of Koo et al. [99] were executed on adult *Arabidopsis* plants, and their results confirmed that the effects of QDs do not only depend on the size and composition but the coat characteristics also have influence on their stability, uptake, and translocation, even in germinating plants. 

### 2.3. The Effect of Dendrimers on Seed Germination 

Dendrimers are hyperbranched nanoscale polymers with a uniform size, described molecular weight, large inner cavities and a high number of surface groups that make them particularly tunable in terms of solution chemistry. Dendritic macromolecules tend to linearly increase in diameter and adopt a more globular shape with increasing dendrimer generation. For this reason, dendrimers have become an ideal way to clearly study the effects of polymer size, charge, and composition on biologically relevant properties, cytotoxicity, blood plasma retention time internalization, biodistribution, and filtration [151]. 

The most common dendrimer scaffold that is commercially available is prepared from polyamidoamine (PAMAM) dendrimers [152]. Dendrimers act as a platform for nitric oxide (NO) transport and delivery, but their application in agriculture is still not explored [153,154]. The usage of dendrimers is widespread, ranging from biology to material sciences [155]. However, NAAS (National Academy of Agricultural Sciences) [156], showed that nanoscale carriers in polymers and dendrimers can be used to efficiently target and deliver pesticides, herbicides, fertilizers or plant growth regulators in plants. Moreover, nanoscale carriers can anchor soil structure and soil organic matter to plant roots [157]. This process can slow down the uptake rate of active ingredients by plant roots, improve compounds stability, reduce the wastes produced, reduce their applied amount, and reduce costs [157,158]. Etxeberria et al. [159] used fluorescent NPs of different composition and sizes and followed their movement into citrus leaves by fluorescent microscopy. Their results indicate that in citrus leaves, the size exclusion limit for NPs is of 5.4 nm. This conclusion was based on the capacity of PAMAM dendrimers G-4 and G-5 (4.5 and 5.4 nm, respectively) to move through the cell wall and into the phloem, but the failure of similar PAMAM dendrimers G-6 (6.7 nm) to move through the apoplast. Dendrimer NPs with the size of 5.4 nm and smaller were observed to penetrate the leaf tissue, and then be taken up and mobilized by the phloem elements. This study provides evidence on the size limit for nanoparticle use in agriculture. The work of Santiago-Morales et al. [160] showed that amine-terminated third-generation (G3) PAMAM dendrimer affects normal seed germination of monocotyledonous and dicotyledonous species, such as *Lolium perenne* (ryegrass), *Lycopersicon esculentum* (tomato) and *Lactuca sativa* (lettuce). The toxicity of G3 PAMAM-(NH_2_)_32_ for seed germination was low, with an EC_10_ in the order of 100 mg L^−1^ (14.5 µM). It is interesting to note that seeds exposed to a low concentration of dendrimer resulted in germination index (GI) above the control, this being particularly intense for *Lactuca sativa*.

Dendrimers find application as delivery agents to carry drugs, oligonucleotides or other molecules for agricultural utility as stated above. Dendrimers are important in increasing the solubility of active agents; improving the adhesion and penetration of active ingredients to plant surfaces; improving the water fastness of an active substance to the plant or seed; increasing the soil penetration of the active agents to reach the plant roots or under soil parts; and reducing the soil adhesion of the active ingredients to reach the plant roots. They can also provide protection against the enzymatic degradation of the active agent by plant or seed or microorganisms present in the soil [161]. 

The number of research data regarding the influencing effect of dendrimers on seed germination and seedling growth is scarce. Therefore, future research should focus on the dendrimer-triggered effects on plants in order to provide the theoretical basis of agricultural applications.

### 2.4. The Effect of Composite Nanomaterials on Seed Germination 

Composite NMs or nanocomposites are hybrid materials produced by the combination of two or more materials with different properties. 

Multiple mechanisms of action are associated with nanocomposites in biological systems, including the disruption of the cell wall and plasma membrane, the inhibition of protein synthesis and DNA replication, and an increase in the oxidation of cell components and compounds [162]. Different nanocomposites are also generally utilized by the industry as antimicrobial agents [163,164], food packaging [165,166], the enhancement of plant physiological parameters [167] and against food oxidation [168]. Nanocomposites are basically divided into three different classes according to their matrix structures: ceramic matrix nanocomposites (Al_2_O_3_/TiO_2_, Al_2_O_3_/SiO_2_, Al_2_O_3_/SiC, Al_2_O_3_/CNT); metal matrix nanocomposites (Co/Cr, Fe-Cr/Al_2_O_3_, Fe-MgO); and polymer matrix nanocomposites (poliester/TiO_2_, polimer/CNT) [162,169,170]. 

Chitosan, polylactic acid and hydroxyl-ethyl methacrylate-based NMs are examples of widely used nanocomposites. These NMs can be prepared easily and faster, with less raw material, unlike conventional polymers. Additionally, composite nanomaterials, which have high stability in biological fluids, have properties such as biodegradability, renewability and biocompatibility [171,172]. Singh et al. [173] studied the effect of nanoscale TiO_2_-activated carbon composite (AC-TiO_2_) on *Solanum lycopersicum* (L.) and *Vigna radiata* (L.) seed germination. Their results showed that the increase in nanocomposite concentration up to a certain level improves the germination rate and reduces the germination time. Accordingly, employing AC-TiO_2_ nanocomposites at a suitable concentration may promote seed germination and also reduce the germination time in *Solanum lycopersicum* and *Vigna radiata*. As reported by Liu et al. [174], NMs could promote germination and rooting early for rice seeds and seedlings and the growth of rice at the tillering stage was obviously affected by nanocomposites. They indicated that the grain yield of rice and nitrogen agronomic utilization efficiency was increased after applying nano-carbon-incorporated SRF (slow release fertilizer). Abdel-Aziz et al. [175] studied how nano chitosan-NPK fertilizer enhanced the growth and productivity of *Triticum aestivum* (wheat) plants grown in sandy soil. Results indicated that all yield variables of wheat plants treated with increasing concentrations of nanocomposite NPK fertilizer (CS-PMAA-NPK (chitosan polymethacrylic acid nanoparticles loaded with nitrogen, phosphorus and potassium)), led to a significant increase in all growth parameters (root length, shoot length, fresh weight, dry weight, water content and leaf area), determined throughout the adult and reproductive growth and developmental stages. 

The ability to develop these properties enables the use of composite materials in a variety of industries, including agriculture, energy, cosmetics and pharmaceuticals [176]. Today, applications of nanocomposites do exist, but the full potential is still not discovered [162]. Therefore, more research is needed to explore the effects of composite nanomaterials on seed germination and plant growth.

## 3. Conclusions and Future Perspectives

In this review, we attempted to emphasize that the increasing worldwide usage of NMs not only provides us technical, medical, industrial, agricultural, cosmetic novelties and conveniences but also creates challenges and possible hazards for all living organisms (including plants). 

Due to excellent reports, we may presume that most of the NMs can enter the plants through the root epidermis, even at the germinative phase, which might be biotransformed into inonic form and translocated to the upper plant parts depending on several characters of the applied NMs such as the size, shape, stability, charge or coating. Since seed germination is the very early and sensitive phase of plant ontogenesis, it is important to determine how different NMs can influence it, especially in the case of crops. 

From the reports overviewed, we may hypothesize that the stimulative effects of various NMs on seed germination are materialized via breaking seed dormancy, increasing germination rate and seedling vigor at the individual level, while at the cellular level, the upregulation of genes involved in cell division, increased antioxidant capacity or reprogrammed carbon/nitrogen metabolism is demonstrated. 

The negative effects of different NMs in growing seedlings, according to the published results, are probably due to inter alia chromosomal aberrations and mitotic abnormalities (e.g., MWCNTs themselves are capable of binding to DNA), and therefore, reduced cell division in the root meristem and root shortening, hormonal imbalance, ROS/RNS overproduction and an increased level of lipid peroxidation, as summarized in Figure 4. 

In agriculture, some kind of seed pre-treatment (i.e., seed priming) seems to be beneficial regarding seed germination efficiency, seedling growth or protection against pathogens, as it has been supported by many experimental data. We may suppose that nanopriming due to the presence of the seed coat generally provides a slow uptake of NMs by the developing plant, which upregulates the antioxidant defense system, resulting in better germination efficiency and plant growth.

We have promising data from germination tests about the dual effects of NMs on developing young plants, but we must take into account that plants usually grow in the soil, exposed to numerous internal and external factors, and thus, further realistic (open air, less controlled) experiments are needed to be carried out in addition to the laboratory studies,

## Figures and Tables

**Figure 1 plants-09-01745-f001:**
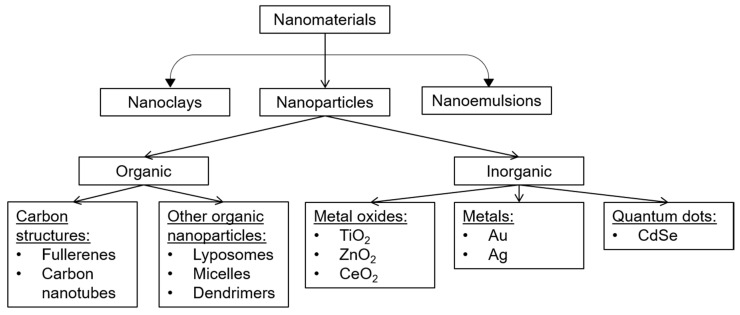
Classification of different nanomaterial (NM) groups (according to [7], with modifications).

**Figure 2 plants-09-01745-f002:**
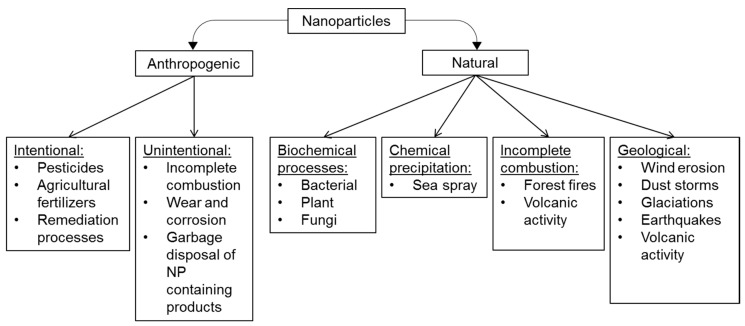
The origin of different nanoparticles according to Buzea és Pacheco [21] with modifications. The abbreviation NP refers to ‘nanoparticle’.

**Figure 3 plants-09-01745-f003:**
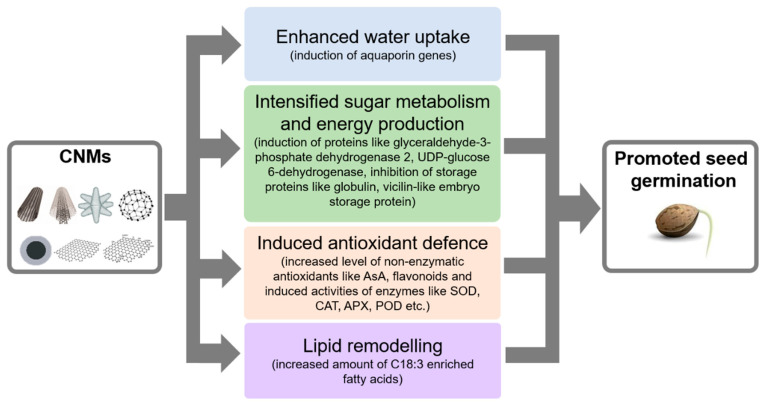
Biochemical and molecular mechanisms of the germination-promoting effect of carbon nanomaterials (CNMs) described so far. Upon CNM exposure, enhanced water uptake, intensified sugar metabolism/energy production, induced antioxidant defense and the remodeling of membrane lipids in seeds have been described in different experimental systems. See details in the text. Abbreviations: AsA, ascorbic acid; SOD, superoxide dismutase; CAT, catalase; APX, ascorbate peroxidase; POD, peroxidases.

**Figure 4 plants-09-01745-f004:**
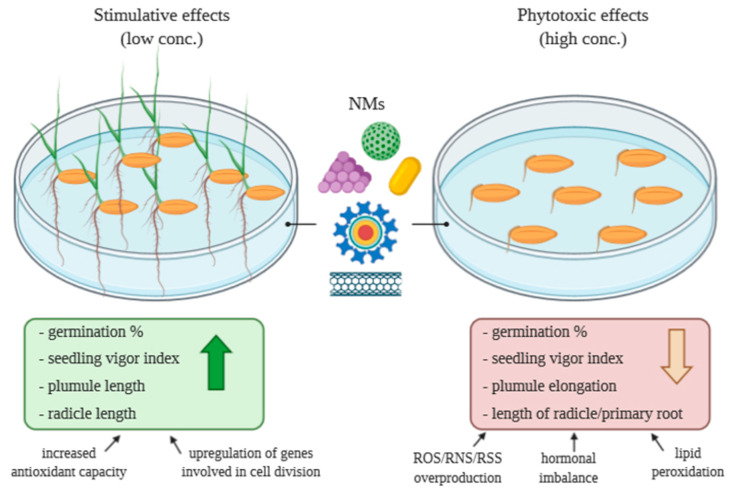
Schematic model summarizing the stimulative or inhibitory pathways of NMs on seed germination (figure created in BioRender.com).

**Table 1 plants-09-01745-t001:** Positive effects of metal and metal oxide NPs on seed germination and seedling growth.

	Plant Name	Size of NM	Type and Chemical Composition of NM *	Duration of Pre-Cultivation	Conc. of the NM Exposure	Time of Exposure	Growth Conditions	Main Effects on Germination and Early Growth **	Reference
**Positive effects**	*Cucumis sativus* L.	10.1 ± 4.2 nm	Au NP	-	62 µg mL^−1^	7 days	Petri dishes (germination test)	Germination index ↑ns	[105]
	*Lactuca sativa* L.							Germination index ↑	
	*Pennisetum glaucum* L.	14–35 nm	Au NP	Seed soaking for 2 h in test solution	20 and 50 µg mL^−1^	5 days	Petri dishes (germination test)	Germination % ↑ns and ↑; total seedling length ↑ns	[106]
	*Cucurbita pepo* L.	20 nm	Ag NP	2 h seed priming	0.05–2.5 mg L^−1^	12–16 days after priming	Petri dishes (germination test)	Germination % ↑ at 0.5–2.0 mg L^−1^ conc.; root length ↑ at 0.05–1.5 mg L^−1^	[107]
	*Citrullus lanatus* L.							Germination % ↑ at 0.5–2.0 mg L^−1^ conc.; root length ↑ at 1–2.5 mg L^−1^	
	*Lolium multiflorum* L.	Width: 122 ± 35 nm, length: 11,908 ± 6703 nm	Ag NW	-	10 ppm	6 days	Petri dishes (germination test)	Root length ↑ns and physical separation from NWs caused further increment	[108]
	*Phaseolus vulgaris* L. ‘Bali’ and ‘Delfina’	~10 nm	Ag NP	Seed priming for 1.5 h	0.25, 1.25 and 2.5 mg L^−1^	5 days	Petri dishes (germination test)	Germination % ↑ at all conc.	[109]
	*Zea mays* L.	<50 nm	Co_3_O_4_	-	269.3–1000 mg kg^−1^ soil (DW)	14 days	Germination test in pot experiment	Germination % ↑ns at all conc.	[110]
	*Alyssum homolocarpum*	10–25 nm	TiO_2_ NP	-	10–80 mg L^−1^	10 days	Petri dishes (germination test)	Germination % ↑ at 10–40 mg L^−1^ conc.	[111]
	*Nigella sativa* L.							Germination % ↑ at 10–40 mg L^−1^ conc.	
	*Salvia mirzayanii* Rech. f. & Esfand							Germination % ↑ at all conc.	
	*Arachis hypogea* L. var. ‘K-134’	25 nm	ZnO NP	100, 1000 and 2000 ppm for seed priming	2 or 30g/15 L for foliar spraying	3 h seed priming then 2x foliar spraying	Pot and field experiment	Germination % ↑, seedling vigour ↑	[112]
	*Capsicum annuum* L.	no data	ZnO NP	6 h seed priming	0.25, 0.5 and 0.75 g	14 days	Moistened blotter paper (in Petri dishes)	Concentration-dependent ↑ of seed germination %, root length ↑, seedling length ↑	[113]
	*Capsicum chinense* L. var. Chichen Itza	18 ± 8 nm	ZnO NP	-	100–500 mg/L	72 h seed priming; 14 day-long germination	Petri dishes (germination test)	Germination % ↑ with conc.; radicule length ↑ at 300 mg/L	[114]
	*Cucumis sativus* L. ‘Poinsett 76’	8 nm	ZnO NP	-	50–1600 mg L^−1^	Until 65 % of the seeds were germinated	Petri dishes (germination test)	Germination % ↑ at 400–1600 mg L^−1^ conc.	[115]
	*Vigna radiata* L.	~18 nm	ZnO NP	Seed priming for 3 h	20, 40, 60, 80 and 100 mg L^−1^	Germinating for 7 days after priming	Petri dishes (germination test)	Germination % ↑	[116]
	*Vigna unguiculata* L.	30 nm	ZnO NP	-	250, 500 and 750 ppm	6 h seed treatment	Soil (pot experiment)	Seedling length ↑, germination % ↑, seedling fresh weight ↑ and vigour index ↑	[117]

* Abbreviations: NC—nanocubes, NP—nanoparticles, NW—nanowires, LR—long nanorods, SR—short nanorods. ** ↑ indicates significant and ↑ns indicates non-significant increase, while ↓ refers to significant decrease and ↓ns to non-significant reduction.

**Table 2 plants-09-01745-t002:** Mixed or no effect of metal and metal oxide NPs on seed germination and seedling growth.

	Plant Name	Size of NM	Type and Chemical Composition of NM *	Duration of Pre-Cultivation	Concentration of the NM Exposure	Time of Exposure	Growth Conditions	Main Effects on Germination and Early Growth **	Reference
**Mixed or no affected**	*Alnus subcordata* L.	no data	Ag NP	-	10–100 mg kg^−1^ (soil)	15 days	Soil (in Petri dishes)	No change of germination % and seedling length	[118]
				-	10 and 20 mg L^−1^		Petri dishes (germination test)	Germination % ↓; seedling length ↓ns and ↓	
	*Lactuca sativa* L.	29.2 ± 1.1 nm	Ag NP	-	100 µg mL^−1^	7 days	Petri dishes (germination test)	Germination index: no change	[105]
	*Pennisetum glaucum* L.	13 nm	Ag NP	Seed soaking for 2 h in test solution	20 and 50 mg L^−1^	5 days	Petri dishes (germination test)	Germination % ↑ at higher conc.; total seedling length ↓ at higher conc.	[119]
	*Pinus sylvestris* L.	no data	Ag NP	-	10–100 mg kg^−1^ (soil)	15 days	Soil (in Petri dishes)	No change of germination % and seedling length	[118]
				-	10 and 20 mg L^−1^		Petri dishes (germination test)	Germination % ↓; seedling length ↓ns and ↓	
	*Triticum aestivum* L. cv. NARC-2009	10–20 nm	Ag NP	-	25–150 ppm	7 days	Petri dishes (germination test)	Germination % ↑ns and number of seminal roots ↑ at 25–75 ppm but ↓ at higher conc.	[120]
	*Zea mays* L.	20 nm	Ag NP	2 h seed priming	0.05–2.5 mg L^−1^	12–16 days	Petri dishes (germination test)	No effect on germination %; root length ↓ at all conc.	[107]
	*Cucumis sativus* L. ‘Poincett 76’	7 nm	CeO_2_ NP	-	500–4000 mg L^−1^	9 days	Petri dishes (germination test)	Germination % ↓ at 2000 mg L^−1^ conc.; root and shoot length ↑ at all conc.	[121]
	*Glycine max* L.	7 nm	CeO_2_ NP	-	500–4000 mg L^−1^	Until 65% of control roots were 5 mm long	Petri dishes (germination test)	Germination % ↓ at 2000 mg L^−1^ conc.; root elongation ↑ at all conc.	[30]
	*Medicago sativa* L. Mesa variety	7 nm	CeO_2_ NP	-	500–4000 mg L^−1^	9 days	Petri dishes (germination test)	Germination % was not affected; root length ↓ at 2000–4000 mg L^−1^, while shoot length ↑ at 500–1000 mg L^−1^ conc.	[121]
	*Zea mays* L.	7 nm	CeO_2_ NP	-	500–4000 mg L^−1^	8 days	Petri dishes (germination test)	Germination % ↓ at 500–2000 mg L^−1^; root length ↑ at 4000 mg L^−1^, while shoot length ↑ at 2000 mg L^−1^ but ↓ at 4000 mg L^−1^ conc.	
	*Brassica oleracea* L.	<50 nm	Co_3_O_4_	-	269.3–1000 mg kg^−1^ soil (DW)	14 days	Germination test in pot experiment	Germination % ↑ at 269–350 mg kg^−1^ conc. but ↓ns at 769–1000 mg kg^−1^conc.	[110]
	*Cicer arietinum* L.	<50 nm	Fe_2_O_3_ NP	Seed priming for 2 h	10–200 mg L^−1^	3 days	Petri dishes (germination test)	Germination time ↓ at all conc.; root length ↓ns but shoot length ↑ns concentration-dependently	[122]
	*Oryza sativa* L. (Y Liangyou 1928)	40–100 nm	Fe_2_O_3_ NC	Seed priming for 2 h	5–150 mg L^−1^	10 d after priming	Petri dishes (germination test)	Germination % ↓ns while root length ↑ and shoot length ↑ns with conc.	[123]
		Length: 200–400 nm, diameter: 10–20 nm	Fe_2_O_3_ SR					Germination % ↓ns; root length ↑ at 5–50 mg/L and ↑ns at 100–150 mg L^−1^; shoot length ↑ and ↑ns at 5–50 and 100 mg L^−1^	
		Length: 500 nm, diameter: 50 nm	Fe_2_O_3_ LR					Germination % ↓ at 5–100 mg L^−1^; root length ↑ with conc.; shoot length ↑ at 5–50 mg L^−1^	
	*Brassica napus* L. var. RGS003	20 nm	TiO_2_ NP	-	10–2000 mg L^−1^	7 days	Petri dishes (germination test)	No change of germination % at 100–1700 mg L^−1^ but ↑ns at 2000 mg L^−1^ conc.; no significant changes of radicle length while plumule length ↓ at 10–1000 mg L^−1^ and ↑ns at higher doses	[124]
	*Allium cepa* L.	21 nm	TiO_2_ NP	-	10–50 mg L^−1^	10 days	Wet filter paper (germination test)	Germination % ↑ns at 10–40 mg L^−1^ conc.; radicle length ↑ns at 10–30 mg L^−1^ but ↓ at higher doses, while shoot length ↑	[125]
	*Carum copticum* L.	10–25 nm	TiO_2_ NP	-	10–80 mg L^−1^	10 days	Petri dishes (germination test)	Germination % ↑ at 10–20 mg L^−1^ but ↓ at higher conc.	[111]
	*Oryza sativa* L. (Y Liangyou 1928)	20 nm	TiO_2_ NP	-	5–150 mg L^−1^	Seed priming for 2 h then cultivation for 10 d	Petri dishes (germination test)	Germination % ↓ns; root length ↑ at 5–10, 50 and 100 mg L^−1^; shoot length ↑ at 150 mg L^−1^	[123]
	*Sinapis alba* L.	10–25 nm	TiO_2_ NP	-	10–80 mg L^−1^	10 d	Petri dishes (germination test)	Germination % ↑ at 10–20 mg L^−1^ but ↓ at higher conc.	[111]
	*Allium cepa* L.	20 nm	ZnO NP	-	10–40 mg L^−1^	10 days	Wet filter paper (germination test)	Germination % and seedling growth ↑ns at lower conc.	[126]
	*Avena sativa* L.	no data	ZnO NP	-	750, 1000 and 1250 mg kg^−1^ seed	10 min priming	Wet paper and field experiment	Germination %, seedling vigour and yield ↑ at low conc., root and shoot length ↓ at higher doses	[127]
	*Brassica juncea* L. Czern. cv. Negro Caballo	45 nm	ZnO NP	-	25 and 100 mg L^−1^	5 days	Petri dishes (germination test)	Primary root length ↑ at 25 mg L^−1^ but ↓ at 100 mg L^−1^ conc.	[32]
	*Brassica napus* L. cv. GK Gabriella			-					
	*Glycine max* L.	8 nm	ZnO NP	-	500–4000 mg L^−1^	Until 65% of control roots were 5 mm long	Petri dishes with wet filter paper (germination test)	Germination was not affected; root elongation ↑ at 500 mg L^−1^ but ↓ at 2000 mg L^−1^ ZnO NP	[30]
	*Hordeum vulgare* L.	30 nm	ZnO NP	-	5, 10, 20, 40 and 80 mg kg^−1^	7 days germination then 21 days cultivation	Petri dishes (germination) then pot experiment	No effect on seed germination and root elongation	[128]
	*Oryza sativa* L.	no data	ZnO NP	1–3 days	10–1000 mg L^−1^	7 days	Moistened filter paper	No change in germination %	[129]
	*Oryza sativa* L.	≤ 50 nm	ZnO NP	2 h	10–500 mg L^−1^	5–12 days	Petri dishes (filter paper or soil)	No change of germination %	[130]
	*Pennisetum glaucum* L.	<50 nm	ZnO NP	-	100–1000 mg L^−1^	7 days	Petri dishes (germination test)	Germination % ↓; root length ↑ but ↓ at 500–1000 mg L^−1^ conc.; shoot length ↑ ns and ↓ns	[131]
	*Phaseolus vulgaris* L. var. red hawk kidney	93.8 or 84.1 nm	ZnO NP	-	62.5–500 mg kg^−1^ (soil)	45 days	Soil (pot experiment)	No effect on germination %	[132]
	*Solanum lycopersicum* L. hybr. ‘tomato cherry super sweet 100’	28 ± 0.7 nm	ZnO NP	Seed priming for 1 h	10–1000 mg L^−1^	5 days	Petri dishes (germination test)	No change of germination % at 10–750 mg L^−1^ but ↓ at 1000 mg L^−1^ conc.	[133]
	*Trifolium alexandrium* L.	no data	ZnO NP	10 min priming	750, 1000 and 1250 mg kg^−1^ seed	no data	Wet paper and field experiment	Germination %, seedling vigour and yield ↑ at low conc., root and shoot length ↓ at higher doses	[127]

* Abbreviations: NC—nanocubes, NP—nanoparticles, NW—nanowires, LR—long nanorods, SR—short nanorods. ** ↑ indicates significant and ↑ns indicates non-significant increase, while ↓ refers to significant decrease and ↓ns to non-significant reduction.

**Table 3 plants-09-01745-t003:** Negative effects of metal and metal oxide NPs on seed germination and seedling growth.

	Plant Name	Size of NM	Type and Chemical Composition of NM *	Duration of Pre-Cultivation	Concentration of the NM Exposure	Time of Exposure	Growth Conditions	Main Effects on Germination and Early Growth **	Reference
**Negative effects**	*Cucumis sativus* L.	29.2 ± 1.1 nm	Ag NP	-	100 µg mL^−1^	7 days	Petri dishes (germination test)	Germination index ↓	[105]
	*Lolium multiflorum* L.	35 ± 7 nm	Ag NP	-	10 ppm	6 days	Petri dishes (germination test)	Root length ↓ns but physical separation from NPs caused ↑	[108]
	*Lolium multiflorum* L.	44 ± 7 nm	Ag NC					Root length ↓ns but physical separation from NCs caused ↑	
	*Phaseolus radiatus* L.	no data	Cu NP	24 h	200–1000 mg L^−1^	48 h	Petri dishes (germination test, agar)	Seedling growth and root growth ↓ concentration-dependently	[135]
	*Triticum aestivum* ssp. aestivum	no data	Cu NP					Seedling growth and root growth ↓ concentration-dependently	
	*Solanum lycopersicum* L. ‘Pomodoro’	7 nm	CeO_2_ NP	-	500–4000 mg L^−1^	6 days	Petri dishes (germination test)	Germination % ↓ns and ↓; root elongation ↓ at 1000–4000 mg/L conc.	[121]
	*Avena sativa* L.	<50 nm	Co_3_O_4_ NP	-	269.3–1000 mg kg^−1^ soil (DW)	14 days	Germination test in pot experiment	Germination % ↓ns at higher conc.	[110]
	*Solanum lycopersicum* L.							Germination % ↓ns but ↓ at 1000 mg/kg conc.	
	*Oryza sativa* L. Jijing No.6.	<50 nm	CuO NP	2 h priming	25–2000 mg L^−1^	Germination for 5 days after priming	Petri dishes (germination test)	Root length ↓ at all conc.	[136]
	*Zea mays* L. Zhengdan No. 958.	<50 nm	CuO NP	2 h priming	25–2000 mg L^−1^	Germination for 7 days after priming	Petri dishes (germination test)	Root length ↓ at all conc.	[136]
	*Arabidopsis thaliana* ‘Col-0’	<50 nm	Fe_3_O_4_ NP	5 days at 4 °C (in dark)	400, 2000 and 4000 mg L^−1^	18 days	1/2 MS medium	Seed germination % ↓ns at 400 and 2000 mg L^−1^; root elongation ↓	[137]
	*Cucumis sativus* L.	7.57 ± 5.6 nm	Fe_3_O_4_ NP	-	116 µg mL^−1^	7 days	Petri dishes (germination test)	Germination index ↓	[105]
	*Lactuca sativa* L.							Germination index ↓	
	*Vicia narbonensis* L.	<100 nm	TiO_2_ NP	Soaking seed for 24 h in the test solutions	0.2–4.0‰	72 h	Petri dishes (germination test)	Germination % ↓ ns; root elongation ↓ ns and ↓, chromosomal aberration index in root tip meristem ↑ with conc.	[138]
	*Zea mays* L.								
	*Arabidopsis thaliana* ‘Col-0’	~44 nm	ZnO NP	5 days at 4 °C (in dark)	400, 2000 and 4000 mg L^−1^	18 days	1/2 MS medium	Seed germination % ↓ and root elongation ↓	[137]
	*Brassica napus* L. cv. Hayola 401	<50 nm	ZnO NP	-	5–500 mg L^−1^	6 days	Petri dishes (germination test)	Germination % ↓ns	[139]
	*Cucumis sativus* L.	≤50 nm	ZnO NP	2 h	10–500 mg/L	5–12 days	Petri dishes (filter paper or soil)	Germination % ↓ns	[130]
	*Oryza sativa* L. Jijing No. 6.	<50 nm	ZnO NP	2 h priming	25–2000 mg/L	Germination for 5 days after priming	Petri dishes (germination test)	Germination % was not affected at 2000 mg L^−1^conc.; root length ↓ at 100–2000 mg L^−1^	[136]
	*Solanum lycopersicum* L. ‘Roma FV’	8 nm	ZnO NP	-	50–1600 mg L^−1^	Until 65 % of the seeds were germinated	Petri dishes (germination test)	Germination % ↓ at 800–1600 mg L^−1^conc.	[115]
	*Solanum lycopersicum* L.	<50 nm	ZnO NP	-	100–1000 mg L^−1^	7 days	Petri dishes (germination test)	Germination % ↓ at 750–1000 mg L^−1^	[131]
	*Triticum aestivum* L.							Germination % ↓ from 250 mg L^−1^ ZnO NP	
	*Triticum aestivum* L.	≤50 nm	ZnO NP	2 h	10–500 mg L^−1^	5–12 days	Petri dishes (filter paper or soil)	Germination % ↓ ns	[130]
	*Vigna radiata* L.								
	*Zea mays* L. Zhengdan No. 958.	<50 nm	ZnO NP	2 h priming	25–2000 mg L^−1^	Germination for 7 days after priming	Petri dishes (germination test)	Germination % was not affected at 2000 mg L^−1^ conc.; root length ↓ at 500–2000 mg L^−1^	[136]

* Abbreviations: NC—nanocubes, NP—nanoparticles, NW—nanowires, LR—long nanorods, SR—short nanorods. ** ↑ indicates significant and ↑ns indicates non-significant increase, while ↓ refers to significant decrease and ↓ns to non-significant reduction.

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
