# Peer review of "Dual Effect of Nanomaterials on Germination and Seedling Growth: Stimulation vs. Phytotoxicity"

_plants, 2020, doi:10.3390/plants9121745_

Round 1

Reviewer 1 Report

The review presented by SZŐLLŐSI et al. is quite interesting as it deals with the impact that different types of nanomaterials have on the germination and growth of seedlings of different crops. Above all, it presents data that show positive impacts (stimulation) and also negative impacts that can be derived from phytotoxicity. However, there are some details that need to be improved to make the manuscript clearer.

Specific comments

Why use the term "priming" in the title of the manuscript? Perhaps it would be better to use the term "stimulation" or something similar.

The introduction is not clearly presents the goal. That aspect should be improved.

A fundamental aspect of the manuscript is the reference to the differences that exist between the types of nanomaterials. However, although a description is presented, the differences between the types of NMs discussed are not clear. It is suggested to expand this section to improve clarity, especially by explaining the physicochemical differences of each type of nanomaterial.

The authors present information that describes the impact that different nanomaterials have on seedling germination and development. However, as the manuscript stands, the advantages or disadvantages of each nanomaterial are not adequately appreciated. It is suggested to include an additional section where the different impacts that nanomaterials can have on the germination and growth of seedlings can be clarified. Above all, emphasizing the mode of action of each nanomaterial.

The conclusions and perspectives section should be improved. The authors present in the first two paragraphs the main findings of the review, however, after there they only refer to carbon-based nanomaterials and dendrimers. 

Author Response

Dear Reviewer,

Thank You for Your comments and recommendations.

Wer corrected the title, as You suggested.

We rewrote the Introduction, but did not give so much plus information about the charecteristics of each type of NM because we think that it is out of focus in review and due to their several variable parameters (features) NMs must be characterized in each experiment (we wrote about it).

 Moreover, we have revised and corrected the Conclusions, too. We hope it will be suitable.

Reviewer 2 Report

I read the manuscript with interest; however I think you have to improve introduction, discussion and the conclusion 

1.In the introduction, author describe lots of NMs in manuscript, but each NMs maybe have different pathway to promote plant germination, the author should describe more detail results or findings about this research topic.

2, The introduction and conclusion parts should be re-write.

3.In line 27-30, the author describe about the hazardment of NMs, I suggest the author show arrange more review information on the toxicity of NMs, even to environment and plants.

4.I suggest the author not to mention about the NMs can be apply to food industry, in recent, less of NMs  can be used in foods, because of its stability, toxicity and price.

5.The pathway of NMs on promote the germination or seedling growth should be discussed.

Author Response

Dear Reviewer,

Thank You for Your comments and suggestions.

We rewrote the Introduction, and we tried to emphasize the promoting effects of each NM type.

Moreover, we made efforts to focus on the pathways, too, as You suggested.

Moreover, we have revised and corrected the Conclusions, and added a summerizing figure about the pathways.

We hope it will be suitable.

Round 2

Reviewer 1 Report

The authors adequately made the suggested changes. Definitely the manuscript improved, so it can be accepted for publication. It is only recommended to check the format of the citations and references as there are some errors.

Author Response

Dear Reviewer, Thank You for your recommendations.

In line 33: after [5] we deleted the „(Piccinno et al., 2012)”

Line 78: We inserted „food package” at the end of the sentence.

We also added the Funding informations and the declaration about the Conflicts of Interest at the end of the text.

We corrected the formal and typographical mistakes in References no. 1., 7., 14., 18., 20., 21., 22., 23., 25., 27-29., 36., 43., 53., 58., 64., 122., 127., 129., 134., 158.
